# Graphite Modified Polylactide (PLA) for 3D Printed (FDM/FFF) Sliding Elements

**DOI:** 10.3390/polym12061250

**Published:** 2020-05-29

**Authors:** Robert E. Przekop, Maciej Kujawa, Wojciech Pawlak, Marta Dobrosielska, Bogna Sztorch, Wojciech Wieleba

**Affiliations:** 1Centre for Advanced Technologies, Adam Mickiewicz University Poznan, 61-614 Poznań, Poland; rprzekop@amu.edu.pl (R.E.P.); bs33013@amu.edu.pl (B.S.); 2Faculty of Mechanical Engineering, Wroclaw University of Science and Technology, 50-370 Wrocław, Poland; maciej.kujawa@pwr.edu.pl (M.K.); wojciech.wieleba@pwr.edu.pl (W.W.); 3Faculty of Materials Science and Engineering, Warsaw University of Technology, 00-661 Warsaw, Poland; m.dobrosielska0000@gmail.com

**Keywords:** PLA, additives, modifiers, friction, wear, properties, sliding elements

## Abstract

With the development of 3D printing technology, there is a need to produce printable materials with improved properties, e.g., sliding properties. In this paper, the authors present the possibilities of producing composites based on biodegradable PLA with the addition of graphite. The team created composites with the following graphite weight contents: 1%, 2.5%, 5%, 7.5%, and 10%. Neat material was also subjected to testing. Tribological, mechanical, and chemical properties of the mentioned materials were examined. Measurements were also made after keeping the samples in ageing and climatic ovens. Furthermore, SEM observations of samples before and after friction tests were carried out. It was demonstrated that increasing graphite content caused a significant decrease in wear (PLA + 10% graphite had a wear rate three times lower than for a neat material). The addition of graphite did not adversely affect most of the other properties, but it ought to be noted that mechanical properties changed significantly. After conditioning in a climatic oven PLA + 10% graphite has (in comparison with neat material) 11% lower fracture stress, 47% lower impact strength, and 21% higher Young’s modulus. It can be certainly stated that the addition of graphite to PLA is a step towards obtaining a material that is low-cost and suitable for printing sliding spare parts.

## 1. Introduction

3D printing technology was initiated in the 1980s. However, its rapid development began about a decade ago. In 2009, the patent for the FDM (fused deposition modelling) printing method expired, and due to this a lot of companies started to offer FDM printers to the industry and individual clients [1].

3D printing gives the possibility of obtaining complex shapes easily, shaping the interior of the object, and it does not create waste during material processing (unlike, e.g., machining). 3D printing is mainly suitable for unit production, obtaining prototypes or elements adapted to highly individual requirements. Nowadays, due to their advantages, 3D printers work in various industries: e.g., electronics [2,3,4], medicine [5,6,7], and the space industry [8].

There are a few common types of 3D printing, depending on the material and printing mechanism used: DLP (Digital Light Processing) [9], SLA (StereoLithography) [10], SLS (Selective Laser Sintering) [11], and FDM [12]. Compared to the others, the FDM method is simple, requires inexpensive devices, and gives good results. Therefore, FDM has gained the greatest popularity (it is estimated that about 60% of devices use FDM technology) [12]. FDM uses thermoplastic polymers as the building material. One of the most commonly used materials in FDM is polylactide, since it adheres well to the printer bed during printing, requires a relatively low temperature of the printer bed and extruder, has a relatively low thermal shrinkage and does not emit harmful substances during printing.

Additionally, a significant feature of polylactide (PLA) is its biodegradability. Environmental pollution and numerous legal regulations have led to increased demand in biomaterials [13]. PLA is one of the main materials on the market of biodegradable polymers [14,15,16]. PLA is a linear aliphatic polyester that is formed by polymerization from lactic acid monomers. Monomers are obtained by fermenting starch from, e.g., maize or sugar beet [17,18]. Despite the fact that polylactide is used in 3D printing, it is also often used as a packaging material. In addition to packaging applications, biopolymers can be used in various industries such as agriculture, automotive, construction, and electronic equipment.

Polylactide, like other biopolymers, has its weaknesses in terms of some properties, e.g., low impact strength, weak gas barrier, poor heat resistance, and low crystallization rate [19,20,21]. Therefore, scientists look for methods of polymers’ modification by introducing a polymer into the matrix, e.g., (nano) fillers, plasticizers, flame retardants [22,23,24,25,26,27,28].

The introduction of additives to PLA used in 3D printing is a standard procedure. However, papers that analyze the impact of various additives on the properties of printed PLA focus on determining the mechanical properties of composites [29,30,31]. No papers describing the influence of additives on PLA tribological properties were found. It is crucial to find a way to improve the PLA tribological properties as this would allow rapid prototyping of low loaded bearings and machine parts exposed to abrasive wear. Moreover, 3D printing would make it simple to print spare sliding parts of machinery and equipment. In case of a failure, it is possible to print a missing component relatively quickly, resume operation of the device and wait for the delivery of a spare part without interrupting production. This possibility is particularly valuable in hard-to-reach areas, e.g., in mines.

One of the additives used to modify PLA is graphite, as it has a low price and excellent thermal and electrical properties. Moreover, it can be used as a filler for the production of polymer composites with competitive multifunctional properties [32]. Moreover, graphite improves the tribological properties of polymeric materials very well. However, publications that concern PLA with graphite addition focus on the influence of graphite on electrical and thermal conductivity [33,34].

The previous research conducted by one of the authors of this publication [35] proved that the addition of graphite improves wear resistance and lowers the coefficient of friction in cooperation with C45 steel. The authors decided to conduct detailed research on PLA-graphite composite in order to find the material with better tribological properties suitable for 3D printing technology (FDM).

## 2. Materials and Methods

The obtained composites were characterized by FT-IR spectroscopy, contact angle analysis (hydrophobic-hydrophilic character), tensile strength test, thermogravimetric analysis (TG), differential scanning calorimetry (DSC), and tribological measurements. During the research, measurements were carried out at six different stations. Therefore, the Materials and Methods section is broken down into subsections to ensure better readability.

### 2.1. Preparation of Samples

Test samples were made using a 3D printer. Since no printer material (filament) in the form of PLA with the addition of graphite is available on the market, the filament was made by the researchers who conducted the study. An amount of 800 g PLA Ingeo™ 2003D was mixed with 200 g graphite with a grain size <40 µm using a ZAMAK MERCATOR WG 150/280 laboratory mill to give a final masterbatch concentration of 20% mentioned above. The process was carried out at 210 °C for 12 min to obtain initial homogeneity of the masterbatch. Then the material was granulated with the WANNER C17.26sv mill. The obtained granulate was mixed with PLA during extrusion of the stream with repeated granulation on a twin-screw extrusion line with a HAAKE Rheomex OS profiled head to obtain final graphite concentrations of 1%, 2.5%, 5%, 7.5%, and 10%. After mixing, granulate was dried for 24 h at 40 °C in a dryer. A 1.75-mm diameter filament was extruded from the obtained granules using a HAAKE Rheomex OS single screw extruder. Extrusion process temperatures were: 170 °C/210 °C/185 °C/170 °C (from the feed zone to the head). Three types of samples were printed from the obtained filament: for bending tests (Figure 1a) for tensile tests (Figure 1b) and tribological tests (Figure 1c). Printing parameters are shown in Table 1.

### 2.2. Methods

#### 2.2.1. Contact Angle Analysis (Hydrophobic-Hydrophilic Character)

Contact angle analyses were performed by the sessile drop technique at room temperature and atmospheric pressure, with a Krüss DSA100 goniometer. Three independent measurements were performed for each sample, each with a 5-µL water drop, and the obtained results were averaged, which was done to reduce the impact of surface nonuniformity.

#### 2.2.2. Colorimetry

Measurements of color change were performed with an EnviSense NR60CP colorimeter with a silicon photodiode detector. An aperture of 4 mm was used. The CIElab method was used. There were three measurements made for each system and the results were averaged.

According to ISO/CIE 11664-4:2019, the CIElab method consists of determining the basic three-chromatic components, which are described as mathematical color models. The CIELab system consists of a and b axes, which are placed at right angles to each other and indicate the color tone. The third L axis indicates brightness.

Parameter *a* indicates color from green (negative values) to red (positive values). The b axis determines the color from blue (negative values) to yellow (positive values). The L parameter is luminance (brightness) and describes the color in the range from black to white. Negative values mean a change to darker, whereas positive values mean a change to lighter. For black, its value is 0, while for white it is 100.

Using the CIELab system, differences between ΔE shades are determined. It is the distance between two points in three-dimensional space, which can be written in the form of dependence:(1)$$\Delta E=\sqrt{\Delta L^{2}+\Delta a^{2}+\Delta b^{2}},$$
where:(2)$$\Delta L=L_{1}-L_{2},$$
(3)$$\Delta a=a_{1}-a_{2}$$
(4)$$\Delta b=b_{1}-b_{2}$$

They respectively mean the difference in the given color parameters between the two samples. The color change is assessed based on the ΔE parameter change (Table 2).

The 1B dumbbell specimens were placed in an ATLAS UV TEST weathering station. The measurement was made according to the norm ISO 4892-3, and the measuring cycle according to ASTM G154: UV-irradiation 0.71 W/m^2^ at 60 °C for 4 h, condensation at 50 °C for 4 h. The full time of measurement was 500 h. The total energy irradiated towards a sample during the whole experiment was 639 kJ/m^2^.

#### 2.2.3. FT-IR Spectroscopy

The Fourier transform infrared (FT-IR) spectra were recorded on a Nicolet iS50 Fourier transform spectrophotometer (Thermo Fisher Scientific) equipped with a diamond ATR unit with a resolution of 0.09 cm^−1^. The spectra were collected in the 400–4000 cm^−1^ range, 16 scans collected for each spectrum.

#### 2.2.4. Flexural and Tensile Strength Tests

For flexural and tensile strength tests, the obtained materials were printed into type 1B dumbbell specimens in accordance with EN ISO 527 and EN ISO 178. The tests of the obtained specimens were performed on a universal testing machine INSTRON 5969 with a maximum load force of 50 kN. The traverse speed for tensile strength measurements was set at 5 mm/min, and for flexural strength at 2 mm/min. Charpy C-notch impact test was performed on an Instron Ceast 9050 impact-machine following ISO 179-1. For all series, 6 measurements were performed.

#### 2.2.5. Thermogravimetric Analysis (Determination of Sample Thermal Stability)

Thermogravimetry was performed using a NETZSCH 209 F1 Libra gravimetric analyzer. The samples of 5 ± 0.2 mg were cut from each granulate and placed in Al_2_O_3_ crucibles. Measurements were conducted under nitrogen (flow of 20 mL/min) in the temperature range of 30–800 °C and a 20 °C/min temperature rise.

#### 2.2.6. DSC Analysis (Determination of Phase Transitions)

Differential scanning calorimetry was performed using a NETZSCH 204 F1 Phoenix calorimeter. The samples of 6 ± 0.2 mg were cut from each granulate and placed in an aluminum crucible with a punched lid. The measurements were performed under nitrogen in the temperature range of −20–290 °C and a 20 °C/min temperature rise. *Tg* was measured for the first heating cycle.

#### 2.2.7. Rheology

The effect of the modifier addition on the mass flow rate (MFR) and volume flow rate (MVR) was also determined. The measurements were taken using an Instron plastometer, model Ceast MF20, according to the applicable standard ISO 1133. The measurement temperature was 190 ± 0.5 °C, while the piston loading was 2.16 kg.

The melt flow index (MFR, MVR) was determined following PN-EN ISO 1133-1: 2011 using an Instron CEAST MF20 weight plastometer. The following test parameters were adopted: load −2.16 kg, temperature −190 °C, procedure-A, standard nozzle (length 8 mm, diameter 2.095 mm). The viscosity measurement was carried out according to the guidelines contained in ISO 11443: 2005 (E) using an Instron Ceast SR 10 capillary rheometer (capillary length—5 mm, capillary diameter—1 mm). The tests were carried out for the shear rate range from 1 s^−1^ to 100 s^−1^.

#### 2.2.8. Microhardness

Shimadzu HMV-2 microhardness tester was used during the research. For the Vickers microhardness measurements, the lowest load (98.07 mN) and the lowest load time (5s) were used.

#### 2.2.9. Tribological Tests

The tribological tests were performed on a pin on disc stand. The test parameters are presented in Table 3 and the position is shown in Figure 2. The wear rate was measured using a micrometer. The height of the sample was checked before cooperation and compared to that obtained after cooperation (dimensions after cooperation were measured only after the temperature of the sample stabilized). Four samples from each material were tested. The value of the friction force was recorded by the computer during the whole test.

#### 2.2.10. Microscopic Observations

The surface imaging was performed with a Quanta FEG 250 (FEI) instrument; SEM at 5 kV and EDS at 30 kV.

## 3. Results

During the research, measurements were carried out at six different stations and the results were followed by microscopic observations. Therefore, the results obtained are broken down into subsections to ensure better readability.

### 3.1. The Contact Angle Analysis (Hydrophobic-Hydrophilic Character)

The contact angle measurements showed that all tested samples are hydrophilic (Table 4). The addition of graphite causes a significant increase in the hydrophobic properties of the obtained composites. As the modifier content increases, the contact angle before the sample ageing process increases. After 250 h of keeping the samples in the ageing oven (exposure to UV radiation, temperature 60 °C), the contact angle slightly decreased. Further exposure of the samples to UV radiation leads to a further decrease in the contact angle for water. The samples with 10% graphite content after 500 h showed absorption of the droplets through the composite surface, which may be due to the agglomeration of graphite in the sample causing damage to the material structure [36]. The exposure of samples in a climatic oven did not lead to such a marked decrease in wettability, and therefore UV radiation ought not to be considered as the main exposure factor in this case.

### 3.2. Colourimetry

Using the CieLab system, measurements of the samples’ color change after exposure to UV radiation for 250 h and 500 h were carried out. The unmodified PLA sample before the ageing oven has L values above 50, which indicates white color (Table 5). The addition of graphite, which is characterized by a dark color, caused a significant decrease in the value of L parameter, which means that the samples have a black color.

Measurements of color change carried out after the UV exposure process in the ageing oven show that *ΔE* is the highest for the base sample after 250 h and 500 h and is 1.97 and 3.64, respectively, which indicates small and medium color changes. For most samples (except for PLA + 2.5% graphite 250 h and 500 h and PLA + 10% graphite 250 h) *ΔE* after the ageing process, for both 250 h and 500 h, is in the range of variation 0 < *ΔE* < 1, which means that there are invisible colour deviations. The addition of graphite increased resistance to color changes of composite materials due to exposure to UV radiation. During the ageing process, polymer decomposition occurs, leading to change in its structure and crystallinity Rodriguez et al. have reported that depolymerization occurring during PLA hydrolysis led to the increase of its crystallinity, which in turn increased opaqueness of the samples [36]. It may be the most significant factor influencing the measured color changes. The degradation of PLA was further proven by FT-IR analysis (see Section 3.3
*FT-IR spectroscopy*).

### 3.3. FT-IR Spectroscopy

FT-IR spectra of PLA with graphite were obtained before and after 250 h and 500 h of photodegradation. All FT-IR spectra except PLA 7.5% 250 h, PLA 7.5% 500 h, PLA 10% 500 h have characteristic bands derived from PLA (Figure 3). The band below 3000 cm^−1^ originates from C–H stretch vibrations. The band at 1760 cm^–1^ corresponds to stretching vibrations from the C=O group. The bands with wave numbers 1450 cm^–1^ and 1360 cm^–1^ come from deformation vibrations CH2 and CH3. Bands from stretching vibrations of C–O bonds occur in the range of 1250–1000 cm^−1^ [37]. Increasing the graphite content in the samples causes band loss, which is caused by the reduced transmittance of a sample itself.

Photodegradation of polylactide under the influence of UV radiation can occur in two ways. The first follows the McNeill and Leiper mechanism [38]. According to the authors, the cause of photodegradation is the C–O–C bond rupture. The second one follows the Norrish type II reaction [39]. The hydrogen atom is transferred to the oxygen atom of the carbonyl group.

The intensity of the characteristic bands for polylactide from the C–H, C=O and C–O–C groups located at 3000–2800 cm^−1^, 1760 cm^−1^, and 1250–1000 cm^−1^, decreases gradually as the samples are exposed to UV radiation [40]. For higher graphite concentration in PLA, i.e., 7.5% and 10%, these bands disappear entirely. It is caused by the total degradation of the polylactide on the surface according to the mechanism of McNeill and Leiper. Photodegradation of polymeric materials occurs the most intensively in the surface layer (up to 10 µm) [41]. This also affects the hydrophobic and hydrophilic properties, the increase in the hydrophilic nature of the samples and the change in their color.

### 3.4. Flexural and Tensile Strength Tests

Strength tests included impact test as well as tensile and bend tests. The tests of breaking strength and bending show that the addition of graphite to a polymer matrix caused an increase in the value of Young’s modulus in relation to the reference sample (Table 6 and Table 7). The highest values were found in the PLA + 10% graphite sample, i.e., with the highest modifier addition. This is due to the high Young’s module of graphite (4100 MPa–27,000 MPa [42]). The particles of graphite interact with the polymer, playing the role of a reinforcing phase, when PLA/graphite composites are subjected to a static load. Up to 7.5% graphite for samples conditioned at 40 °C and up to 5% for samples conditioned in a climate oven, fracture stress and fracture strain were virtually unchanged within the levels of standard deviation. For higher loadings, the strain at the breaking of the modified samples was lower than that of the reference sample, which is due to low tensile fracture strain of the graphite itself, which is below 0.5% [43]. Therefore, graphite particles embedded in PLA either undergo fracture before the failure of the sample itself or they have too little adhesion to the matrix and get pulled out of the material during sample straining, failing to serve a role of a reinforcing filler, especially at high loadings. Both suggestions are supported by decreased fracture stress values. Similar results were obtained by Muriaru et al. [32]. For samples conditioned in a climate oven, the temperature and humidity cycles affected the materials, introducing additional defects, resulting in a further decrease in mechanical parameters, which can also be seen as a disruption of data trend for bending tests.

Bending strength tests showed a gradual decrease in the maximum bending stress for the samples conditioned at 40 °C. The lowest bending stress, similarly to the results of the tensile strength tests, is characterized by a sample with 10% filling, and the difference from the base sample is 12.6 MPa (reduction by 18%). The decrease in the maximum bending stress value for increasing the graphite sample filling is associated with the inevitable formation of agglomerates and poorer dispersion of the filler in the matrix at high loadings, as well as the limited filler-matrix interaction and mechanical properties of the filler itself, as discussed earlier.

The breaking strength and bending tests were also carried out after conditioning the samples in a climatic oven (Figure 4, Figure 5, Figure 6 and Figure 7). A visible trend is an increase in the deformation value of the samples compared to the results before exposure to moisture and temperature (exception the sample PLA + 7.5% graphite), which indicates an increase in the material elasticity. This is due to water being absorbed by the material, which means that water molecules disperse between polymer chains, and reduce chain–chain interaction. It was reported that PLA might absorb up to ~1% water by weight [44]. The largest deformation difference occurs for the reference sample, which indicates the highest water absorption. The values of Young’s modulus parameter and the pre-and post-conditioning stresses are slightly different, which means that these materials are more resistant to weather conditions.

The strength tests carried out indicate that the addition of graphite increased Young’s modulus while reducing the maximum bending and fracture stress. The obtained results agree with the literature data [32].

Bending tests were conducted following the ISO standard 178:2006. Six bars from each series were used. Maximum deflection measured according to the previously mentioned standard was 6 mm. Width of the lower measuring jaw was 64 mm (measuring length). Speed was 2 mm/min. The samples fractured during the bending strength test. 

Samples were fractured during the impact test (fracture type: C). The impact tests show that the addition of graphite to a polymer matrix caused an increase in the KC value in comparison to the reference sample (exception PLA sample + 10% graphite). The samples with 1% and 2.5% filling have the highest KC, and the difference to pure PLA is 1.93 kJ/m^2^ and 1.68 kJ/m^2^, respectively (Table 8 and Figure 8). Composite materials are characterized by higher mechanical strength and greater stiffness, as evidenced by the higher energy needed to break the sample. An exception is a 10 weight of filling with graphite, which KC is 9.74 kJ/m^2^, i.e., it is lower by 0.36 kJ/m^2^ than PLA samples without additives, which is caused by a too high concentration of the modifier, and thus, inferior dispersion. Deterioration of mechanical properties for this sample is also confirmed by bending and tensile tests.

Impact tests were also carried out for materials conditioned in a climatic oven. KC parameter values are higher after exposure of the samples to moisture and temperature. The highest difference was observed for pure PLA and it is 9.98 kJ/m^2^. As for the samples with filling, the difference is in the range of 0.82–2.11 kJ m^2^, which indicates that the modifiers caused water absorption of the composite material to decrease with respect to the reference sample.

### 3.5. Thermogravimetric Analysis (Determination of Sample Thermal Stability)

Graphite has excellent thermal stability, as well as the high thermal conductivity of 3000 Wm^−1^ K^−1^, therefore, as a nanofiller, it can affect the thermal properties of composite materials. The thermogravimetric analysis determined the temperature at 1% weight loss. The addition of graphite to a polymer matrix caused an increase in temperature at 1% weight loss for all modified samples (Table 9). The highest differences in relation to the reference sample were recorded for samples with 2.5% and 5% graphite filling and they are respectively 9.2 °C and 14.3 °C higher. 

The “onset” was also determined from the thermogram, which is the lowest for a pure PLA sample, while for samples with a filler it is in the range of 352.2–354.7 °C. The temperature at the maximum mass change rate is highest for pure PLA, the addition of graphite caused a slight drop in temperature. The addition of graphite to PLA leads to composites with improved thermal stability. Graphite can lead to a delay in thermodegradation of the PLA matrix, which is attributed to the protective effect given by the nanofiller [32]. The addition of graphite reduces the temperature of material decomposition, at the same time, the onset of decomposition (Onset) occurs at a higher temperature. This is due to the increase in the specific surface of the degrading polymer, defects on the surface of the filler contribute to the faster decomposition of the material, similar observations were made in [45] where the polyester matrix composites were modified with the addition of graphene oxide and graphite.

### 3.6. DSC Analysis (Determination of Phase Transitions)

The analysis of differential calorimetry allowed to determine the phase changes occurring in the samples. For both pure PLA and the modified sample, the crystallization temperature was determined. The crystallization temperature is lower for samples with graphite added compared to the reference sample, which confirms the nucleating effect produced by the modifier [32] (Table 10). Higher crystallization peaks occur for composite materials compared to the sample without filler. They decrease with increasing filler concentration, which is caused by poorer dispersion for higher graphite concentrations in the matrix. 

Due to the presence of both crystalline and amorphous forms in PLA, glass transition temperature (amorphous fraction) and softening temperature (crystalline fraction) can be determined from the DSC curve [46].

The glass transition temperature is higher for composite materials with the addition of graphite (exception sample with 5% filling), which indicates an increase in PLA stiffness. This phenomenon is also confirmed by strength tests.

The softening temperature of pure PLA and modified samples is within 152.4–153.3 °C. It is the highest for a sample without filler, which means that pure PLA sample has more crystalline fraction compared to modified samples.

The changes occur with a simultaneous decrease in phase change (crystallization) temperature. There is a visible impact of graphite at low loadings on the crystallization temperature. Such changes cannot be considered insignificant. The fact of crystallinity levels being unaffected at high loadings of graphite can be explained by its agglomeration. For graphite, the critical factor causing polymer chain separation is the lamellar structure of graphite [47].

### 3.7. Rheology

Rheological tests have shown that samples modified with graphite are characterized by lower MFR and MVR flow rates compared to a sample of unmodified PLA (Table 11 and Figure 9). The lowest values occur for samples with 1% and 10% filling, while for 2.5%, 5%, and 7.5% samples the values are similar and are in the range of 5.169–5.855 g/10 min (MFR), 4.169–4.772 cm³/10 min (MVR). Melt flow rates take lower values, which may result from deterioration of plastic flow in the plastometer nozzle due to the increase in flow resistance associated with the solid filler [48]. 

The MFR value is an indication of the material’s behavior during flow in the flow channels. The higher the value is, the longer the flow routes can be used. Materials with a low melt flow rate can be processed by extrusion, while materials with a larger one by extrusion blow-molding method. Viscosity measurements showed that the addition of graphite reduces viscosity (Figure 10). PLA with additives had a higher viscosity than unmodified PLA only when the shear rate was very low (1 s^−1^) Rheology in graphite/PLA system is different than that typical for polyolefins. A distinct decrease in viscosity observed with an increase of graphite loading can be explained based on low interaction between the filler and the matrix, as well as due to microtribological properties of lamellar-structured graphite [49].

### 3.8. Microhardness

Microhardness measurements of PLA samples were carried out before (Table 12 and Figure 11) and after friction tests (Table 13 and Figure 12). The addition of graphite to PLA does not significantly change the microhardness. In most cases, the microhardness changes the amount several percent. It is noteworthy that the microhardness value obtained after friction for a sample with a 10% graphite is 17% lower than that registered for unmodified PLA. It is possible that with an increased proportion of graphite during friction on the PLA sample, a thin layer of graphite has formed and it is responsible for reducing the microhardness. Authors who analyzed friction and wear of composites with graphite described the formation of this layer [50,51,52,53]. A significant reduction in microhardness was observed only for PLA with 10% graphite. Therefore, it appears that a graphite content of 10% is required to create a graphite film on the surface of the sample.

### 3.9. Tribological Tests

The results of the PLA pair wear test with C45 steel (Table 14 and Figure 13) indicate that the addition of graphite affects the course of PLA wear process. Only 1% mass share of graphite reduced *I_h_* wear rate by 26% (relative to unmodified material wear rate). As the graphite content increases to 7.5%, wear rate slightly changes. Only 10% mass share of graphite causes a significant reduction in wear rate (wear rate is lower by 65% compared to the wear rate of unmodified PLA).

The fact that wear rate is significantly lower for a sample with 10% graphite leads to the same conclusion as to when analyzing microhardness measurements. It appears that a graphite content of 10% is required to create a graphite film on the surface of the sample. The formation of graphite film is considered a key factor in reducing wear rate [50,51,52,53].

The results of friction tests in the pair PLA-C45 steel indicate that the addition of graphite to PLA has little impact on the value of friction coefficient *µ* (Table 15 and Figure 14). The decrease in the coefficient of friction against pair with unmodified PLA is up to 11% (with a 1% mass share of graphite). Increasing the proportion of graphite reduces the difference in the coefficient of friction and at 10% the value is the same as for a pair with unmodified PLA.

### 3.10. Microscopic Observations

After the metal-polymer friction tests had been completed, PLA samples were observed using an SEM microscope (Figure 15). It turned out that as the graphite content in a composite increased, more “crumbly, spreadable” flakes appeared on the surface of the sample after friction. For PLA with 7.5% of graphite, not only flakes were observed, but also spread material. A lot of spread material arranged parallel to the direction of cooperation was observed on the surface of the sample with a content of 10% graphite. It appears that a layer containing compressed and spread wear products appeared on the surface. Considering the conclusion about the wear rate and microhardness results, it can be assumed that this layer is a film with a large proportion of graphite.

## 4. Discussion

The assessment of the influence of additives on PLA properties ought to consider two aspects. Firstly, the impact of the additive on the properties of the finished product resulting from 3D printing should be assessed. Secondly, the effect of the additive on the material’s behavior during the printing process ought to be analyzed.

### 4.1. The Influence of Graphite Addition on 3D Printed Part’s Tribological Properties

The main purpose of adding graphite to PLA was to improve the tribological properties of 3D prints. The collected data has shown that prints made of PLA with the addition of graphite were characterized by significantly lower wear rate. A ten percent mass proportion of graphite resulted in a 65% reduction in wear rate compared to unmodified PLA. 

The most commonly used composites based on polymer materials during cooperation with steel have a friction *µ* coefficient in the range from 0.1 to 0.6 and a wear rate *I_h_* from only a few µm/km [54]. Unmodified PLA turned out to be a material that can be assessed as poor in terms of use in sliding elements of machinery and equipment (I_h_ = 32 µm/km, *µ* = 0.45). The ten percent mass addition of graphite to PLA resulted in a reduction in wear rate to a level of about 10 µm/km, which can be considered satisfactory. The addition of graphite at ten percent did not reduce the coefficient of friction, but this is not particularly problematic.

As for the cooperation between plastic and metal, a layer of transferred polymer material on the metal surface is formed [54]. Due to this effect, no lubricant is needed for this type of pair. The possibility of maintenance-free operation and the ecological nature of the solution have determined the suitability of plastics in sliding elements. As a result of friction, the polymer surface changes as well.

Microscopic observations (carried out after friction) of the surface of the PLA sample with the addition of ten percent graphite showed that bands arranged parallelly to the direction of cooperation appeared on the surface. They formed a specific layer on the polymer surface. A significant reduction in wear rate for PLA with a ten percent graphite addition should probably be explained by the high proportion of graphite in this layer because graphite has excellent lubricating properties. It is noteworthy that the surface of the PLA sample after friction had 17% less microhardness. Probably it was caused by the graphite, which is characterized by low hardness.

In the case of friction in pair, the polymer material, metal, is of key importance: mechanical and adhesive [54]. The mechanical component is associated with “hooking” uneven material surfaces together. The adhesive component is an effect of the molecular interaction between the materials. Graphite has excellent lubricating properties and additionally fills surface irregularities, creating a smooth surface. The smoothed surface reduces the value of the mechanical component. What is more, as shown in the conducted tests, the addition of graphite to PLA causes an increase in the contact angle, and thus a weakening of the adhesive interactions between the polymer material and steel.

### 4.2. The Influence of Graphite Addition on the 3D Printing Process

During 3D FDM/FFF printing, the first layer of the material must adhere to the printer bed throughout the entire printing process. It is also vital for the material to be easily extruded through the nozzle. To evaluate the effect of the additive on the behavior of the material during the printing process, the following results were used: a contact angle analysis, determination of sample thermal stability, and a flow coefficients measurements.

The contact angle measurement helps to assess the adhesion between the polymer material and the printer bed. The addition of graphite increases the contact angle, therefore the adhesion will be reduced and problems with maintaining the print on the bed might arise. To determine if this problem is significant, further research ought to be carried out.

During processing plastic in 3D printing using the FDM/FFF method, two temperatures are of the greatest importance: glass transition temperature T_g_ and melting temperature T_m_. The printer bed ought to be heated to a temperature slightly higher than the glass transition temperature of the processed material. Increasing the bed temperature above T_g_ leads to a reduction of the surface tension and a larger contact area between the 3D printer’s bed and the printing material [55]. As a result, better adhesion between the bed and the filament can be noticed. The addition of ten percent graphite slightly increased T_g_ (from about 60 to 63 °C). Therefore, when printing PLA with the addition of graphite, an increased bed temperature should be used. Another material very popular in 3D printing, ABS, requires heating the bed to over 100 °C. Therefore, most of the printers available on the market have the option of heating the table to over 100 °C. The requirement to increase the table temperature by a few degrees Celsius in the range of 60–70 °C will not pose any problems.

As for T_m_, the addition of graphite slightly increased the temperature to which the polymer material can be heated without causing its degradation. Filament manufacturers usually specify the range of recommended extrusion temperature. However, the exact value depends on the 3D printer model and the printing parameters selected by the user. Since PLA with graphite is slightly more resistant to temperature, the printer user will be able to set the extrusion temperature, which works well when printing from PLA without additives.

The values of flow rates were significantly reduced when ten percent graphite was added to PLA. Melt flow rates are crucial when the material is to be injected into molds with long channels. The use of PLA is mainly seen in 3D printing of sliding elements. The channel through which the heated material is extruded in the extruder of the 3D printer has a small length and reduced flow rates should not be a problem. During samples’ printing for the described tests (even for PLA with the addition of ten percent graphite) there were no significant difficulties. However, to ensure that the addition of graphite does not cause difficulties when processing the filament on a 3D printer, additional research should be conducted.

## 5. Conclusions

1. The ten percent mass addition of graphite significantly improved PLA’s wear resistance. However, the wear resistance of PLA with a ten percent graphite mass addition is lower than for other polymer-based composites used for sliding elements, and the friction coefficient is higher. 3D printed sliding elements will serve as prototypes or replacements, therefore their lifespan will be short. Consequently, PLA wear resistance with ten percent mass addition of graphite ought to be assessed as satisfactory. It can be stated with certainty that the addition of graphite to PLA is a step towards obtaining a material which is both low-cost and suitable for printing sliding spare parts.

2. The addition of graphite did not affect the value of the friction coefficient. Considering that the printed sliding elements are meant to work for a short time, low energy consumption in the friction node is not a critical feature. The friction coefficient remains at an acceptable level for short-term operation.

3. The other problem with the use of PLA for sliding elements is the fact that this material is characterized by very poor thermal resistance (even among plastics), which significantly limits the speed of cooperation and pressure when working with steel. Even though graphite has excellent thermal conductivity, the addition of graphite did not increase the softening point of PLA. By adding graphite to PLA, wear resistance was significantly improved; however, the pressure and speed limitations could not be overcome.

4. An important aspect is PLA susceptibility to biodegradation. Sliding replacement parts are expected to be used shortly and these components ought to not pose a threat to the environment. Graphite is practically neutral for the environment, so its addition will not negate the benefits of PLA’s biodegradability.

5. The graphite in PLA reduced the wear rate; however, it also caused undesirable effects. Strength and impact strength were reduced, while the material became more rigid. It is worth mentioning that the sliding elements of polymer materials are not loaded with significant forces. This is because polymer materials have (compared to metals) a low softening temperature. Therefore, they can work with metals only at limited pressure and speed, which will not cause excessive heat. The slight reduction in the strength of the polymer material is not very important, because these materials in sliding elements are still loaded many times with less than permissible forces.

6. The addition of graphite lowered the flow indexes, increased the Tg temperature and the contact angle. Therefore, it is highly possible that for PLA with graphite, optimal printing parameters will be slightly different than for PLA without additives. However, it appears that the requirements for printing PLA with graphite would not differ significantly from those for PLA without additives. However, determining the optimal 3D printing parameters for PLA with graphite requires further research.

## 6. Patents

Patent application: WIPO ST 10/C PL433343.

## Figures and Tables

**Figure 1 polymers-12-01250-f001:**
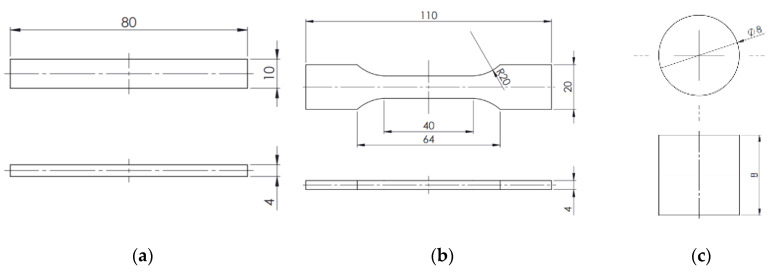
Dimensions of samples used for testing: (**a**) bending; (**b**) stretching; (**c**) tribological. All dimensions are in millimeters (mm).

**Figure 2 polymers-12-01250-f002:**
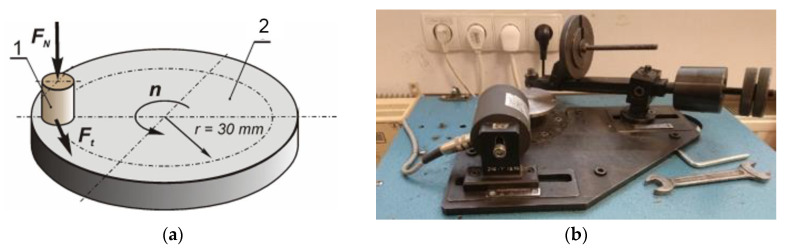
Tribological measurements: (**a**) scheme of tribometer, *Fn*—normal load, *Ft*—friction force, *n*—rotational speed, 1—specimen, 2—disc; (**b**) Pin on disc apparatus.

**Figure 3 polymers-12-01250-f003:**
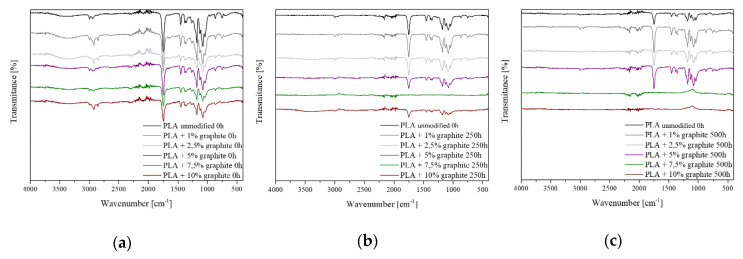
FT-IR spectra for samples: (**a**) non-aged; (**b**) aged for 250 h; (**c**) aged for 500 h.

**Figure 4 polymers-12-01250-f004:**
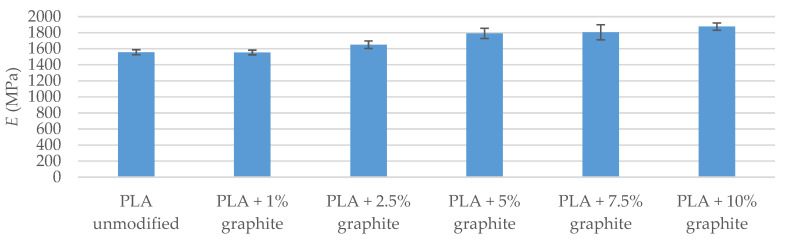
Young’s modulus of stretched samples (after conditioning in a climatic oven).

**Figure 5 polymers-12-01250-f005:**
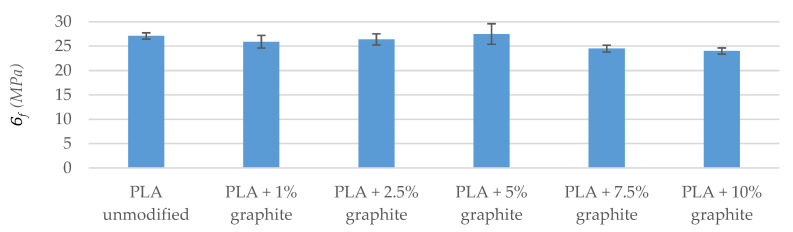
Stress at stretched samples breaking (after conditioning in a climatic oven).

**Figure 6 polymers-12-01250-f006:**
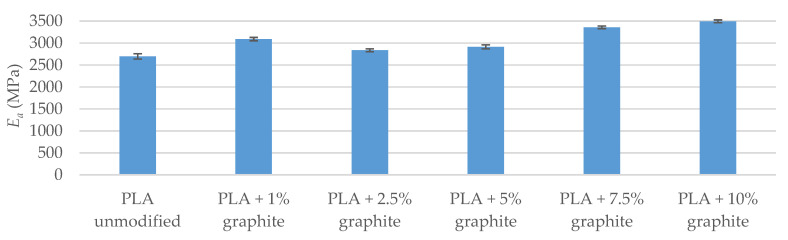
The automatic module of bending samples (after conditioning in the climate oven).

**Figure 7 polymers-12-01250-f007:**
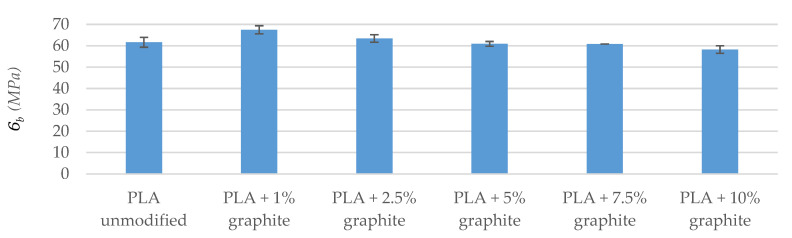
Maximum stress when bending samples (after conditioning in a climatic oven).

**Figure 8 polymers-12-01250-f008:**
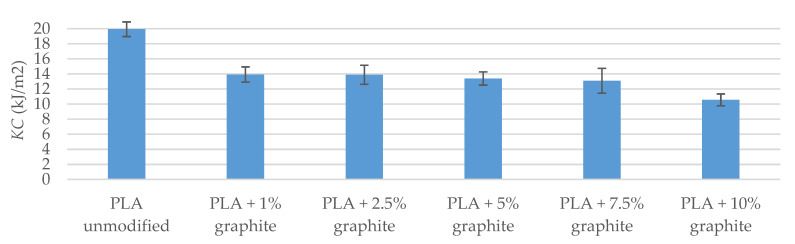
Impact test results (after conditioning in a climatic oven).

**Figure 9 polymers-12-01250-f009:**
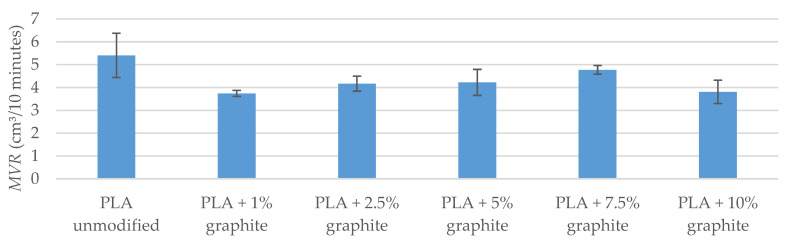
MVR flow factor measurement results.

**Figure 10 polymers-12-01250-f010:**
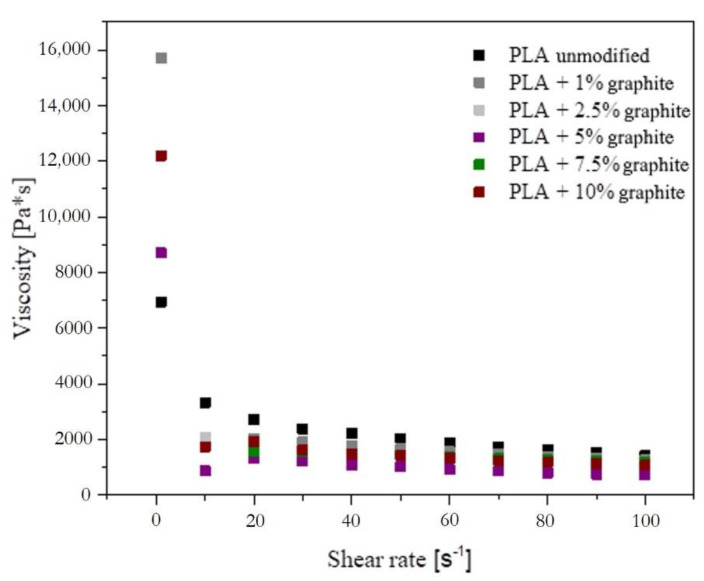
Viscosity measurement results.

**Figure 11 polymers-12-01250-f011:**
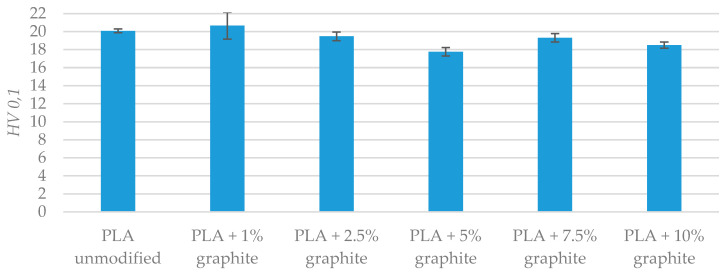
Microhardness measurements’ results before friction test.

**Figure 12 polymers-12-01250-f012:**
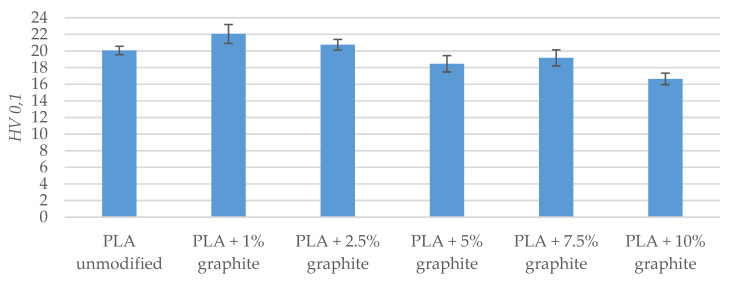
Microhardness measurements’ results after the friction test.

**Figure 13 polymers-12-01250-f013:**
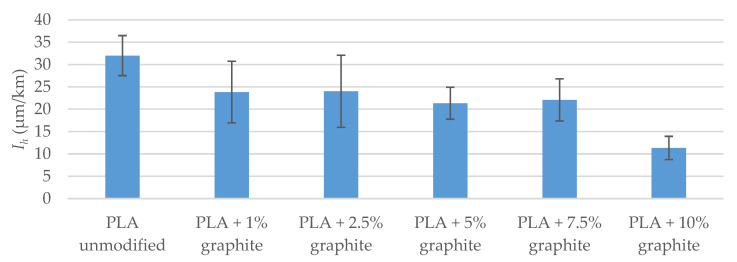
Results of PLA wear tests cooperating with C45 steel (v = 0.34 m/s, and T_0_ = 23 °C, p = 0.11 MPa, Ra = 0.35–0.45 µm).

**Figure 14 polymers-12-01250-f014:**
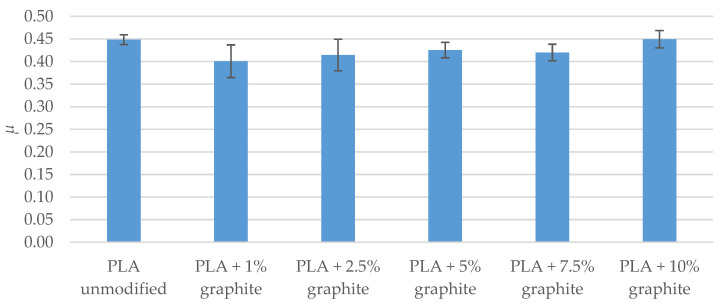
Results of friction tests in PLA pair-C45 steel (v = 0.34 m/s, T_0_ = 23 °C, p = 0.11 MPa, Ra = 0.35–0.45 µm).

**Figure 15 polymers-12-01250-f015:**
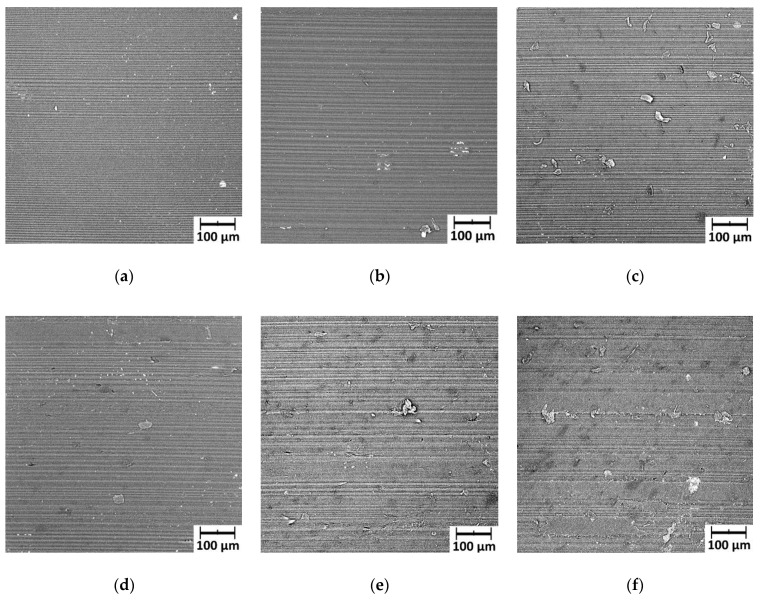
Surface of polymer samples after cooperation with steel: (**a**) PLA unmodified, (**b**) PLA + 1% graphite, (**c**) PLA + 2.5% graphite, (**d**) PLA + 5% graphite, (**e**) PLA + 7.5 % graphite, (**f**) PLA + 10% graphite.

**Table 1 polymers-12-01250-t001:** Process parameters that were requested when printing samples.

	Shape of a Sample and a Bar	Pin
Nozzle diameter	0.4 mm	0.3 mm
Print temperature	200 °C	200 °C
Bed temperature	60 °C	60 °C
Layer height	0.18 mm	0.10 mm
Number of full top/bottom layers	3/3	16/16
Number of contours	2	3
Top and bottom layers style	Rectilinear (parallel to the long edge)	Concentric
Infill style	Honeycomb	Rectilinear
Infill percentage	30%	16%
Cooling	100%	100%
Print speed	40 mm/s	30 mm/s

**Table 2 polymers-12-01250-t002:** Criteria for assessing the change in color based on the change of the ΔE parameter.

Intervals of Variation	A Recognizable Color Difference
0 < ΔE < 1	Invisible color deviations
1 < ΔE < 2	Minimal color differences, recognizable only by an experienced observer
2 < ΔE < 3.5	Average color deviations, recognizable by an inexperienced observer
3.5 < ΔE < 5	Visible color deviations
ΔE > 5	Big differences in color

**Table 3 polymers-12-01250-t003:** Parameters characterizing the conducted friction tests.

Parameter	Value
Pin force on the disc	5.49 N
Pin pressure on the disc–*p*	0.11 MPa
Disc material	Steel C45
Disc roughness	Ra = 0.35–0.45 µm
Duration of test	2 h 27 min
Linear speed in association–*v* Ambient temperature-*T*_0_	0.34 m/s 23 °C
Friction path	3 km
Type of friction	Technically dry

**Table 4 polymers-12-01250-t004:** Contact angle measurements.

	Keeping Samples in the Ageing Oven			Samples Conditioning in a Climatic Oven
	0 h	250 h	500 h	
PLA unmodified	55.3°	56.8°	64.9°	77.8°
PLA + 1% graphite	62.4°	62.1°	63.8°	63.3°
PLA + 2.5% graphite	63.4°	62.0°	62.5°	69.8°
PLA + 5% graphite	57.0°	56.6°	44.3°	78.8°
PLA + 7.5% graphite	72.1°	66.5°	64.9°	55.1°
PLA + 10% graphite	79.8°	65.3°	0°-hydrophilic	77.7°

**Table 5 polymers-12-01250-t005:** Results of colorimetric measurements (ΔE markings: X—invisible color deviations, L—very low color differences, M—medium color deviations).

	0 h		250 h		*ΔE*	500 h		*ΔE*
PLA unmodified	**L**	52.26	**L**	50.91	1.97 (L)	**L**	53.72	3.64 (M)
	**a**	−0.85	**a**	−0.47		**a**	−0.95	
	**b**	1.14	**b**	−0.25		**b**	−2.19	
PLA + 1% graphite	**L**	27.02	**L**	26.66	0.58 (X)	**L**	26.29	0.88 (X)
	**a**	−0.85	**a**	−0.75		**a**	−0.61	
	**b**	−1.61	**b**	−1.17		**b**	−1.17	
PLA + 2.5% graphite	**L**	28.80	**L**	27.40	1.67 (L)	**L**	27.79	1.08 (L)
	**a**	−0.73	**a**	−0.45		**a**	−0.62	
	**b**	−1.73	**b**	−0.86		**b**	−1.37	
PLA + 5% graphite	**L**	29.14	**L**	28.59	0.82 (X)	**L**	28.88	0.34 (X)
	**a**	−0.60	**a**	−0.43		**a**	−0.52	
	**b**	−1.60	**b**	−1.01		**b**	−1.39	
PLA + 7.5% graphite	**L**	28.09	**L**	28.16	0.71 (X)	**L**	28.13	0.39 (X)
	**a**	−0.69	**a**	−0.49		**a**	−0.52	
	**b**	−1.63	**b**	−0.95		**b**	−1.28	
PLA + 10% graphite	**L**	28.22	**L**	27.06	1.33 (L)	**L**	27.70	0.56 (X)
	**a**	−0.70	**a**	−0.52		**a**	−0.64	
	**b**	−1.52	**b**	−0.90		**b**	−1.33	

**Table 6 polymers-12-01250-t006:** Results of tensile tests.

After Keeping the Sample for 24 h at 40 °C				After Conditioning in a Climate Oven		
	Young’s Modulus [MPa]	Fracture Stress [MPa]	Fracture Strain [%]	Young’s Modulus [MPa]	Fracture Stress [MPa]	Fracture Strain [%]
PLA unmodified	1482.4 ± 43.45	25.5 ± 0.70	2.152 ± 0.063	1556.4 ± 31.68	27.1 ± 0.64	2.417 ± 0.097
PLA + 1% graphite	1509.1 ± 22.50	24.4 ± 0.59	2.108 ± 0.030	1553.4 ± 29.01	25.9 ± 1.30	2.273 ± 0.1063
PLA + 2.5% graphite	1706.9 ± 26.09	26.5 ± 0.64	2.108 ± 0.032	1650.1 ± 47.10	26.4 ± 1.14	2.308 ± 0.078
PLA + 5% graphite	1834.0 ± 52.89	26.0 ± 0.86	1.993 ± 0.028	1791.8 ± 63.47	27.5 ± 2.10	2.228 ± 0.130
PLA + 7.5% graphite	1644.5 ± 53.83	26.7 ± 0.43	2.208 ± 0.029	1804.8 ± 94.61	24.5 ± 0.70	2.184 ± 0.115
PLA + 10% graphite	1886.1 ± 47.40	23.8 ± 0.72	1.882 ± 0.039	1875.6 ± 45.36	24.0 ± 0.64	1.952 ± 0.063

**Table 7 polymers-12-01250-t007:** Results of bending tests.

	After Keeping the Sample for 24 h at 40 °C		After Conditioning in a Climate Oven	
	Automatic Module [MPa]	Maximum Bending Stress [MPa]	Automatic Module [MPa]	Maximum Bending Stress [MPa]
PLA unmodified	2857.9 ± 40.41	68.6 ± 1.34	2696.0 ± 61.22	61.6 ± 2.30
PLA + 1% graphite	2835.0 ± 61.21	65.0 ± 1.17	3089.1 ± 38.14	67.5 ± 1.87
PLA + 2.5% graphite	2953.2 ± 47.13	63.9 ± 0.91	2835.8 ± 33.12	63.4 ± 1.77
PLA + 5% graphite	3262.3 ± 38.12	64.2 ± 1.47	2911.9 ± 45.17	60.9 ± 1.12
PLA + 7.5% graphite	3268.0 ± 21.33	60.6 ± 2.02	3358.4 ± 29.47	60.8 ± 0.97
PLA + 10% graphite	3332.1 ± 22.14	56.0 ± 0.98	3493.3 ± 32.18	58.2 ± 1.76

**Table 8 polymers-12-01250-t008:** Impact test results.

	PLA Unmodified	PLA + 1% Graphite	PLA + 2.5% Graphite	PLA + 5% Graphite	PLA + 7.5% Graphite	PLA + 10% Graphite
KC [kJ/m^2^] samples after 24 h in 40 °C	10.10 ± 1.11	12.03 ± 0.77	11.78 ± 1.23	11.45 ± 0.98	11.73 ± 1.94	9.74 ± 0.47
KC [kJ/m^2^] conditioning in a climatic oven	19.93 ± 0.97	13.92 ± 1.01	13.89 ± 1.27	13.39 ± 0.87	13.09 ± 1.64	10.56 ± 0.78

**Table 9 polymers-12-01250-t009:** Results of thermogravimetric analysis.

	PLA Unmodified	PLA + 1% Graphite	PLA + 2.5% Graphite	PLA + 5% Graphite	PLA + 7.5% Graphite	PLA + 10% Graphite
1% of weight loss [°C]	305.8	311.2	315.0	320.1	310.7	309.1
Onset [°C]	348.3	354.6	354.7	353.5	352.6	352.2
Temperature at maximum weight change rate [°C]	376.0	373.1	372.9	375.1	371.6	372.4

**Table 10 polymers-12-01250-t010:** Results of differential scanning calorimetry (DSC) analysis.

	PLA Unmodified	PLA + 1% Graphite	PLA + 2.5% Graphite	PLA + 5% Graphite	PLA + 7.5% Graphite	PLA + 10% Graphite
Glass transition temperature [°C]	59.9	62.5	62.6	57.6	62.7	63.0
Crystallization temperature [°C]	126.4	122.2	120.9	119.2	124.5	125.3
Softening temperature [°C]	153.3	152.6	152.5	152.4	153.0	152.8

**Table 11 polymers-12-01250-t011:** Results of flow coefficient measurements (tests carried out at 190 °C).

	PLA Unmodified	PLA + 1% Graphite	PLA + 2.5% Graphite	PLA + 5% Graphite	PLA + 7.5% Graphite	PLA + 10% Graphite
MVR [cm³/10 min]	5.407 ± 0.97	3.741 ± 0.13	4.169 ± 0.33	4.224 ± 0.57	4.772 ± 0.19	3.807 ± 0.51
MFR [gram/10 min]	6.705 ± 0.79	4.639 ± 0.47	5.169 ± 0.54	5.238 ± 0.34	5.855 ± 0.24	4.721 ± 0.65

**Table 12 polymers-12-01250-t012:** Microhardness measurements’ results before the friction test.

	PLA Unmodified	PLA + 1% Graphite	PLA + 2.5% Graphite	PLA + 5% Graphite	PLA + 7.5% Graphite	PLA + 10% Graphite
*HV 0.1*	20.07 ± 0.21	20.65 ± 1.49	19.48 ± 0.48	17.76 ± 0.46	19.31 ± 0.47	18.49 ± 0.34
Δ*HV 0.1*	-	3%	−3%	−12%	−4%	−8%

**Table 13 polymers-12-01250-t013:** Microhardness measurements’ results after the friction test.

	PLA Unmodified	PLA + 1% Graphite	PLA + 2.5% Graphite	PLA + 5% Graphite	PLA + 7.5% Graphite	PLA + 10% Graphite
*HV 0.1*	20.06 ± 0.51	22.05 ± 1.13	20.75 ± 0.64	18.46 ± 0.98	19.17 ± 0.97	16.64 ± 0.69
Δ*HV 0.1*	-	9%	3%	−8%	−4%	−17%

**Table 14 polymers-12-01250-t014:** Results of PLA wear tests cooperating with C45 steel (v = 0.34 m/s, T_0_ = 23 °C, p = 0.11 MPa, Ra = 0.35–0.45 µm).

	PLA Unmodified	PLA + 1% Graphite	PLA + 2.5% Graphite	PLA + 5% Graphite	PLA + 7.5% Graphite	PLA + 10% Graphite
*I_h_* (µm/km)	32 ± 4	24 ± 7	24 ± 8	21 ± 4	22 ± 5	11 ± 3
Δ*I_h_*	-	−26%	−25%	−33%	−31%	−65%

**Table 15 polymers-12-01250-t015:** Results of friction tests in PLA pair—C45 steel (v = 0.34 m/s, T_0_ = 23 °C, p = 0.11 MPa, Ra = 0.35–0.45 µm).

	PLA Unmodified	PLA + 1% Graphite	PLA + 2.5% Graphite	PLA + 5% Graphite	PLA + 7.5% Graphite	PLA + 10% Graphite
*µ*	0.448 ± 0.011	0.401 ± 0.036	0.414 ± 0.035	0.425 ± 0.017	0.420 ± 0.018	0.449 ± 0.019
Δ*µ*	-	−11%	−8%	−5%	−6%	0%

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
