# Peer review of "Graphite Modified Polylactide (PLA) for 3D Printed (FDM/FFF) Sliding Elements"

_polymers, 2020, doi:10.3390/polym12061250_

Round 1

Reviewer 1 Report

The work is interesting and proposes a new material of immediate practical applicability.
The wealth of experimental data provided makes the work certainly useful for the scientific community and worthy of publication.

Besides, the paper is well written, clear and pleasant to read. Above all, it is rich in data and very informative, all the arguments are supported by experimental evidence.
Conclusions highlight both the advantages and the limitations in the use of the proposed material and on the basis of both suggest a possible scenario of use, represented by the manufacture of elements suitable for sliding contact with metal surfaces; such application is evaluated through the tribological study of the surfaces after the experimental test of sliding phenomena.

Author Response

The manuscript has been improved, language errors corrected. Thank you for your revision.

Reviewer 2 Report

Abstract

Line 20. It is % in weight or volume?

Line 21: ‘Reference material without a modifier’ is a long term to call the reference material. Use neat material, or something similar.

Line 26: Same than in Line 21

Line 27: Show specific data of the decrease in the mechanical properties? What % decrease after the graphite addition? At least at highest graphite amount.

Introduction

Line 46. Maybe it would be interesting to cite this article “Lignin: A Biopolymer from Forestry Biomass for Biocomposites and 3D Printing”.

Line 38: Why authors define PLA as polylactide and not like Polylactic acid? Use the full name or the abbreviatoin PLA

In general, the introduction section is short and not relevant. The 50% of this section is about the explanation of PLA definition and properties. I strongly suggest that the authors define in the introduction the interest of using graphite in materials (specially PLA) and why this type of properties are of interest for 3d printing. Show novelty.

Material and methods

Line 51: Authors define that 800 g of PLA were mixed with 200 g of graphite to five af inal concentration of 20% mentioned above. Where is it amount mentioned? Authors always show that the % amount of Graphite on PLA are 1%, 2.5%, 5, 7.5% and 10%.

Line 58: Diluted is not the correct word. Maybe melted.

Table 3: Replace 5,49 and 0,11 (5.49 and 0.11)

Results

Section 3.1.

 I think that the data of 0° for PLA+10% graphite is not correct. The addition of graphite slight increase the water contact angle and the hidrofobicity of samples, specially for 10% graphite. Why this angle is so high at 0h and 250h and decrease in totally for 500h?

Compare with other works. Is it a normal behaviour? What effect have on the final material properties?

Section 3.2.

What happens during the aging process? Why colour changes? Some oxidation chemical reactions?

Section 3.3.

Cite literature for the peak identification.

Section 3.4.

Why graphite addition produces an increase in some properties? How it interacts with the PLA matrix?

Compare data with litarature. Is it increase enough in comparison with others fillers with similar function? Discuse data.

Section 3.5.

What consequences have the incrase in the thermal stability? Is it a good behaviour for 3 printed, or is harmful? Discuss data.

Section 3.6.

Change the title. Authors repeat 3.5 number in section titles.

Discuss data.

Section 3.7 (Change, author repeat 3.5 again)

Section 3.8 (Change, author repeat 3.5 again)

Section 3.9 (Change, author name it 3.6)

Section 3.10 9 (Change, author repeat 3.6 again)

In general terms, I miss discussion of data. It is like a report and not is comapred with literature. Only 20 references, where 12 is in introduction. The manuscript need more discussion of the data obtained and its relationship with the optimal properties of 3d printed materials and final application.

Reviewer 3 Report

The authors have presented an interesting area of research. The biodegradable PLA is of current demand in 3D printing of polymers components. A wide range of characterizations ranging from physical, mechanical to tribological performance has been performed using the relevant characterization technical. However, the manuscript does not justify the title of the present work. The reviewer has the following major concern.

  1. The introduction needs major revision including the results from the literature. The introduction does not even mention the need for the presented work. The objective of the manuscript is not well defined. 
  2. The English sentences need to modify in order to clearly understand.  For example "800g PLA Ingeno ............ " line 54. 
  3. The materials and methods section needs to change appropriately. It is very difficult to follow.
  4. Fig. 1, what is the unit of the dimensions. Mention either in Fig or caption.
  5. Results are not clearly described and discussed. In contact angle results, the Authors claim that with the increase in %graphite contact angle increased. From the data, this claim is inconsistent (at 5% graphite a dip can be observed, please explain). Also, sample 10% graphite content after 500h, water absorption leading to graphite agglomeration is hypothesized. What is the basis for it? Add reference or give experimental images. In all the cases, the contact angle change is not consistent (not increased/ decreased) with increased %graphite. Please explain in detail.
  6. The inconsistency is also observed with colorimetric data. For example, significant change with the addition of 1% graphite but remain constant with the further addition of graphite. Explain the results in detail.
  7. Explain FTIR data in more detail. It does not give a clear outcome of the data. What is a change in the FTIR spectrum and how does it affects the properties of modified PLA.
  8. what is the meaning of the statement " in the range of 4100 MPa÷ 27000 MPa". Please explain. If it is not correct, change throughout the manuscript.
  9. Not enough discussion is made regarding the mechanical properties of graphite modified PLA. It is claimed that the strain reduces with %graphite which is further related to increased stiffness" It's not necessary that increasing stiffness reduces the fracture strain. In addition, I do not see significant changes in strain and fracture stress.
  10. Be consistent in explaining the results in the TEXT. In the table, it Fractures stress and in the text breaking stress. This makes the reader confused. 
  11. Increased young modulus is not discussed properly. Why change in youngs modulus? Explain in detail.
  12. Why fracture toughness increased initially up to 7.5% graphite (remain the same) and reduced for 10% graphite? 
  13. TGA and DSC data does not essentially show significant improvement after graphite modification of PLA. Please explain.
  14. Rheology property seems improved with graphite. Please explain based on graphite composition.
  15. In hardness data, the reduction in hardness has been attributed to the formation of a thin graphite layer. Please refer to the literature or give experimental evidence.
  16. The Title needs to modify in order to accommodate the scope of the presented work. The present title does not justify the work.
  17. Add section "concussion/summary" of the work.
  18. The reference is not cited appropriately. All the work is before 2018. also, the number of references is very few and does not address all the work on PLA.
  19. Overall a major revision on technical presentation is required. 

Round 2

Reviewer 2 Report

The authors ralized the suggestions made from reviewers

Author Response

Thank you. We've corrected some language errors and numeration in the manuscript.

Reviewer 3 Report

The authors have responded to the reviewer's comments adequately. However, there are a few concerns.

  1. The reviewer still feels that the Title of the manuscript should be changed to include the Tribological property which is the objective of the manuscript
  2. There are many mistakes in spelling. Also, there are mistakes in the numbering of Tables and Figures, for example, Instead of Table 13 and Figure 12, it is wrongly mentioned in Table 11 and Figure 13 (Line. 417) Please make sure the numbering in the manuscript is correct.
